# Glucagon Receptor Signaling and Glucagon Resistance

**DOI:** 10.3390/ijms20133314

**Published:** 2019-07-05

**Authors:** Lina Janah, Sasha Kjeldsen, Katrine D. Galsgaard, Marie Winther-Sørensen, Elena Stojanovska, Jens Pedersen, Filip K. Knop, Jens J. Holst, Nicolai J. Wewer Albrechtsen

**Affiliations:** 1Department of Biomedical Sciences, Faculty of Health and Medical Sciences, University of Copenhagen, 2200 Copenhagen, Denmark; 2Novo Nordisk Foundation Center for Basic Metabolic Research, Faculty of Health and Medical Sciences, University of Copenhagen, 2200 Copenhagen, Denmark; 3Department of Cardiology, Nephrology and Endocrinology, Nordsjællands Hospital Hillerød, University of Copenhagen, 3400 Hillerød, Denmark; 4Center for Clinical Metabolic Research, Gentofte Hospital, University of Copenhagen, 2900 Hellerup, Denmark; 5Department of Clinical Medicine, Faculty of Health and Medical Sciences, University of Copenhagen, 2200 Copenhagen, Denmark; 6Steno Diabetes Center Copenhagen, 2820 Gentofte, Denmark; 7Department of Clinical Biochemistry, Rigshospitalet, 2100 Copenhagen, Denmark; 8Novo Nordisk Foundation Center for Protein Research, Faculty of Health and Medical Sciences, University of Copenhagen, 2100 Copenhagen, Denmark

**Keywords:** alpha cell, amino acids, diabetes, glucose, hyperaminoacidemia, hyperglucagonemia, liver–alpha cell axis

## Abstract

Hundred years after the discovery of glucagon, its biology remains enigmatic. Accurate measurement of glucagon has been essential for uncovering its pathological hypersecretion that underlies various metabolic diseases including not only diabetes and liver diseases but also cancers (glucagonomas). The suggested key role of glucagon in the development of diabetes has been termed the bihormonal hypothesis. However, studying tissue-specific knockout of the glucagon receptor has revealed that the physiological role of glucagon may extend beyond blood-glucose regulation. Decades ago, animal and human studies reported an important role of glucagon in amino acid metabolism through ureagenesis. Using modern technologies such as metabolomic profiling, knowledge about the effects of glucagon on amino acid metabolism has been expanded and the mechanisms involved further delineated. Glucagon receptor antagonists have indirectly put focus on glucagon’s potential role in lipid metabolism, as individuals treated with these antagonists showed dyslipidemia and increased hepatic fat. One emerging field in glucagon biology now seems to include the concept of hepatic glucagon resistance. Here, we discuss the roles of glucagon in glucose homeostasis, amino acid metabolism, and lipid metabolism and present speculations on the molecular pathways causing and associating with postulated hepatic glucagon resistance.

## 1. Introduction

In the 1920s, Kimball and Murlin reported the existence of a pancreatic factor other than insulin with effects on glucose homeostasis (GLUCose-AGONist [1]) [2]. This factor was later identified as glucagon (denoted glucagon 1–29) [3]. Glucagon is a peptide of 29 amino acids with a variety of biological actions including, but not limited to, glucose homeostasis [4,5,6]. A recent PubMed search on glucagon revealed that more than 45,000 articles relating to glucagon have been published, and extensive efforts are currently being made to further study and understand the physiological role(s) of glucagon [4,7].

The biology of glucagon is intriguing not only from a physiological perspective but indeed also from a therapeutic point [1,8,9]. Glucagon receptor antagonism reduces blood glucose in patients with type 2 diabetes, and glucagon co-agonism (with, e.g., an incretin hormone) reduces body weight in overweight type 2 diabetes patients [10]. The complexity of glucagon´ suggested diabetogenic effect is greater than anticipated, as antagonizing its actions affects fasting but not postprandial blood glucose levels [11]. This is in contrast to what one may have suspected, as the ‘hyperglucagonemia’ observed in some type 2 diabetes patients has been reported to be pronounced in the postprandial state (the lack of glucose-induced suppression of alpha cell secretion, which we will come back to later) [12,13,14]. The impact of glucagon therefore differs between fasting and prandial conditions, at least when it comes to glucose homeostasis. Investigating the physiological actions of glucagon on not only glucose homeostasis but, importantly, also on lipid and amino acid metabolism is needed for understanding glucagon biology at fasting and prandial conditions. Specifically, biased (meaning activation of specific intracellular pathways) targeting of glucagon receptor signaling [15] towards, e.g., hepatic glycogenolysis or gluconeogenesis may potentially represent an ‘intelligent pharmacological’ approach exploiting, e.g., the glucose lowering effects of glucagon receptor antagonism without its undesired effect on hepatic lipid metabolism (increased liver fat [16]).

The purpose of this review is to assess the key aspects of glucagon receptor signaling not only regarding glucose homeostasis but also with respect to amino acid and lipid metabolism. We will also discuss the evidence and potential relevance of the new concept of hepatic ‘glucagon resistance’. We begin this review by introducing the production, secretion, and measurement of glucagon before turning towards glucagon receptor signaling in regards to glucose, amino acid, and lipid metabolism.

## 2. Processing of Proglucagon

The *GCG* gene, encoding the glucagon precursor proglucagon, is well conserved across species [17]. Proglucagon has 160 amino acids and is expressed in certain neurons of the brain stem, in intestinal L cells, and in pancreatic alpha cells [17]. Several bioactive peptides, including glucagon-like peptide 1 (GLP-1) and glucagon-like peptide 2 (GLP-2), are cleaved from proglucagon by prohormone convertase(s) in a tissue-specific (or perhaps more accurately enzyme-specific [18]) manner (Figure 1). The differential processing of proglucagon appears to reflect the enzymatic activities of the two prohormone convertases: prohormone convertase 1/3 (PC1/3) and 2 (PC2) [19]. Proglucagon therefore gives rise to a variety of peptides. Thus, throughout the small and large intestine, proglucagon-producing cells termed L cells are located within the epithelium [20,21] in an ideal position to sense the variety of nutrients and microbial products and convey the information to the rest of the body via the secretion of GLP-1, GLP-2, oxyntomodulin, and glicentin, which contribute to the regulation of appetite, bone resorption, gastrointestinal growth, and glucose homeostasis [1,22,23,24,25]. With co-expression of PC1/3 (e.g., in intestinal L cells), proglucagon is cleaved to form glicentin, oxyntomodulin, GLP-1, and GLP-2; whereas with PC2 expression as in the alpha cells, proglucagon is cleaved to form mainly glucagon and the so-called major proglucagon fragment [26,27,28]. In line with this, mice deficient of PC1/3 are incapable of producing GLP-1, while mice deficient of PC2 cannot produce glucagon [29,30,31,32].

The absolute selectivity of PC1/3 and PC2 remains a matter of discussion. It has been speculated that metabolic stressors such as type 2 diabetes, obesity, and Roux-en-Y gastric bypass surgery may alter the processing profile of proglucagon both in the pancreas and in the gut, but the extent to which this occurs in humans and the clinical relevance of such changes remain unknown [18].

## 3. Secretion of Glucagon

Glucagon is secreted in response to a variety of metabolic signals [6,33] such as changes in blood glucose concentrations [2,34], certain amino acids [35], perhaps free fatty acids [36], and in response to stress [37] (e.g., activation of the sympathetic nervous system). Here, we shortly discuss some of the currently suggested mechanisms underlying glucose-dependent glucagon secretion. For further insight, please see Ref. [4,38,39,40].

In humans, blood glucose levels are reciprocally correlated to glucagon secretion, and the potential intrinsic glucose-sensing mechanisms of alpha cells have been studied for decades using a variety of techniques [6,41,42,43]. As an example, the physiological roles of sodium and potassium channels have been studied in whole islets and isolated alpha cells using electrophysiological techniques (patch clamping). Paracrine factors also play an important role and some have argued that combining or integrating intrinsic and paracrine factors is needed to uncover the enigmatic mechanism of glucose-induced inhibition of glucagon secretion [44,45].

The mechanisms underlying glucose-induced inhibition of alpha cell secretion are still a matter of debate. One of the proposed intrinsic pathways leading to hypoglycemia-induced glucagon secretion is a decrease in the ATP/ADP ratio, which paradoxically slightly increases K_ATP_ channel activity, leading to voltage-dependent increased activity of P/Q type calcium channels and a subsequent influx of Ca^2+^ [46]. In vivo, another important mechanism may be hypoglycemia-induced activity of the pancreatic sympathetic innervation [47]. The potent regulation of glucagon secretion by glucose from isolated perfused pancreas preparations supports a direct effect of hypoglycemia on the alpha cell [48], unless stimulatory neurons of the intra-pancreatic ganglia are glucose sensitive.

Hyperglycemia-induced attenuation of glucagon secretion may be even more complex due to extra-pancreatic signals such as the glucagonostatic effect of GLP-1 or the glucagonotropic effects of GLP-2 and glucose-dependent insulinotropic polypeptide (GIP), all three of which are secreted in response to the ingestion of carbohydrates. In addition, paracrine signals elicited by glucose, in particular delta cell-secreted somatostatin and beta cell-secreted insulin, may also inhibit glucagon secretion. A recently published study demonstrates that glucokinase, expressed in alpha cells, may play an important role in glucose-regulated glucagon secretion [49,50]. The inhibitory effect of glucose is preserved in whole islets but appears to be lost when alpha cells are isolated and studied selectively [51]. This again points to an important effect of intra-islet factors secreted from the neighboring beta and delta cells. Somatostatin secreted from the pancreatic delta cells and insulin secreted from pancreatic beta cells in response to increased glucose concentrations both inhibit glucagon secretion [52], although this may depend on the experimental setting [53]. Other intra-islet factors have also been implied in the regulation of glucagon secretion: urocortin 3 [54], zinc [55], GABA/L-glutamate [56], gamma-aminobutyric acid [57], amylin [58], and ephrin [59]. Furthermore, several extra-islet signals have been reported to contribute to the regulation of glucagon secretion and include, but are not limited to, GLP-1 [60], GIP [61], GLP-2 [62], ghrelin [63], and gastrin [64]. Finally, the newer class of glucose-lowering drugs for the treatment of type 2 diabetes, sodium-glucose co-transporter 2 (SGLT-2) inhibitors, have been associated with increased plasma glucagon concentrations. The absolute differences in plasma glucagon concentrations between SGLT-2 inhibitor-treated and placebo-treated patients are small (1–1.5 pmol/L) [65,66], and although some have argued for a direct role of SGLT-2 on the alpha cells [67,68,69,70], current evidence, which includes the results of glucose clamping, seems to point toward an indirect mechanism by which SGLT-2 inhibition increases renal glucosuria, resulting in significant lowering of blood glucose concentrations that causes a slightly greater alpha cell secretion [71,72,73,74,75]. Whether or not increased plasma glucagon after SGLT-2 inhibition has any clinical impact is currently unknown, but some may argue that increasing hepatic glucose production when combined with reduced glucose reuptake in the kidneys may limit the reduction in blood glucose (and hence clinical efficacy).

Clearly then, changes in glucagon secretion induced by glucose, for instance, depend on the experimental conditions and may differ dramatically between experiments in isolated alpha cells, isolated islets, isolated perfused pancreas preparations, and in vivo. Intra- and extra-islet factors implicated in the regulation of glucagon secretion are summarized in Figure 2.

## 4. Accurate Measurement of Glucagon

A major limitation to the study of glucagon physiology and pathophysiology has been the hitherto limited accuracy of many immunoassays used to measure glucagon [76]. Some studies have applied additional analytical techniques such as size-exclusion chromatography and mass-spectrometry for characterization of the measured moiety. Falsely increased glucagon immunoreactivity may be due to degradation of circulating glucagon-containing molecules such as glicentin, oxyntomodulin (in vivo as well as post sampling), or proglucagon with resulting formation of (immunoreactive) glucagon. Another explanation may be insufficient specificity, e.g., cross-reactivity of the antibody/antibodies used in the assay with peptides with related sequences to that of glucagon [77,78]. The term hyperglucagonemia may therefore not in all cases relate to increased plasma concentrations of bioactive glucagon but may as well be ascribed to preanalytical and analytical challenges. The analytical and physiological importance of N-terminally elongated glucagon-like moieties is uncertain [78,79,80], but part of the concentration of immunoreactive glucagon measured in plasma with the single antibody C-terminal immunoassays may certainly be due to such molecular forms, in particular proglucagon 1–61 [79]. This may be particularly important in patients with impaired kidney function [81].

## 5. Glucagon Receptor Signaling

### 5.1. Glucagon and Glucose Homeostasis

A fundamental aspect of glucagon signaling is its role as a regulator of glucose homeostasis [43,82]. Increased plasma glucagon levels result in increased hepatic glucose production [83]. Therefore, glucagon is traditionally known as a counter-regulatory hormone to the hypoglycemic effects of insulin [84]. This balance of glucagon and insulin signaling is thus largely responsible for maintaining physiological euglycemia [85]. In hypoglycemic conditions, glucagon secretion increases, which results in increased hepatic glucose production via a number of cellular mechanisms involving suppression of glycogenesis and glycolysis and stimulation of glycogenolysis and gluconeogenesis [86,87] (Figure 3).

Once glucagon binds its seven transmembrane receptor on the plasma membrane of the cell, it leads to conformational changes that activate G_αs_-coupled proteins [83]. This consequently increases cAMP levels via the activation of adenylate cyclase, which in turn stimulates activation of protein kinase A (PKA) and cAMP response element-binding (CREB) protein [87]. CREB is responsible for inducing transcription of glucose 6-phosphatase and phosphoenolpyruvate carboxykinase (PEPCK), both of which contribute to increased gluconeogenesis. Meanwhile, PKA activation results in a number of intracellular events in addition to phosphorylation of CREB [41,86]. Firstly, it phosphorylates (hence activates) the phospho-fructokinase 2 (PFK-2)/fructose 2,6-bisphosphatase (FBPase2) protein. Upon phosphorylation, the PFK-2 activity is inhibited while the FBPase2 activity is activated, leading to increased fructose 6-phosphate levels. This induces gluconeogenesis, while decreased fructose 2,6 bisphosphate levels result in reduced glycolysis [41,86]. Secondly, PKA activates pyruvate kinase, which increases fructose 1,6 bisphosphate levels and thus lower pyruvate levels, which suppresses glycolysis [88]. Thirdly, PKA activates phosphorylase kinase, which, via activated phosphorylase, increases conversion of glycogen to glucose 1-phosphate in addition to inhibiting glycogen synthase, which overall lowers glycogen levels and results in hepatic glucose release [89].

During periods of short-term fasting (less than ~12 h), glucose levels in the bloodstream rely mainly on glycogenolysis, but over time as glycogen stores become depleted, the maintenance of euglycemia increasingly depends on gluconeogenesis [89]. Because the effect of glucagon on gluconeogenesis relies on gene transcription, the physiological effects do not result until hours after glucagon secretion, whereas the stimulatory effect of glucagon on glycogenolysis has observable effects within minutes due to the rapid phosphorylation cascade [90]. Given that gluconeogenesis is a substrate-dependent mechanism and glucagon does not feed gluconeogenesis via provision of, e.g., muscle-derived amino acids (to date no glucagon receptor has been identified on human skeletal cells), other mechanisms, including catecholamines and cortisol [91,92,93], must act in order to maintain blood glucose during long-term fasting. However, glucagon stimulates hepatic amino acid metabolism [94], resulting in an increased flux of amino acids into the hepatocytes and is thereby also able to provide substrates in the form of gluconeogenic amino acids. The relative contribution of glucagon to glucose production therefore varies depending on glycogen content and the availability of gluconeogenic substrates in the circulation. This is in concordance with studies showing that during prolonged fasting (when glycogen stores are depleted, and the hepatic glucose production depends on gluconeogenesis), the effect of glucagon on hepatic glucose production is weak [83,91,92]. Glucagon may therefore be ineffective in regards to hepatic glucose production when glycogen stores are exhausted, e.g., during starvation and prolonged exercise [84,88].

This is exemplified by the findings of attenuated glucose production in response to exogenous glucagon after one week of low carbohydrate diet (LCD) compared to the response after a week of high carbohydrate diet (HCD) in patients with insulin pump-treated type 1 diabetes [95,96]. The lowered glucose production in response to glucagon is likely to be due to the reduced storage of hepatic glycogen in the LCD group [97,98]. However, acute glucagon administration after a period of LCD caused a significantly higher increase in free fatty acid (FFA) and ketone body plasma concentrations compared to glucagon administration after the HCD. This indicates that the LCD may have promoted the use of fat over carbohydrates, resulting in increased fat oxidation and ketogenesis [99], supporting the conclusion that glucagon regulates glucose homeostasis mainly through glycogenolysis rather than gluconeogenesis. These results suggest that diet carbohydrate content must be accounted for when treating hypoglycemia with low-dose glucagon, as the amount of glycogen is crucial for the effect of glucagon on blood glucose concentrations [100].

The above-mentioned studies are of clinical interest as glucagon is used to treat hypoglycemia—in most cases due to exogenous hyperinsulinemia (iatrogenic). Furthermore, low dose glucagon injection is currently being investigated in connection with the development of dual-pumps (glucagon + insulin infusion) [101,102,103,104,105]. Recent studies suggest that the amount of glucose produced (assumable by hepatic glucose production) after glucagon injection may not be influenced by body weight—in the same study, insulin levels across patients and testing time points remained constant. Therefore, the capacity for glucagon to appropriately regulate plasma glucose concentrations may, as suggested, not be affected by body weight and hence the existence of glucagon resistance (see the Section regarding glucagon resistance). In contrast, studies in animal models of hepatic steatosis have reported an attenuated effect of glucagon on glucose production thereby possibly linking pathophysiological traits of the liver to glucagon resistance [106,107]. Preliminary data from our group did, however, not support the notion of glucagon resistance with respect to its ability to stimulate hepatic glucose production in humans, but in contrast revealed an impaired effect on amino acid metabolism [108].

As alluded to above, it is poorly understood how glucagon secretion is suppressed during hyperglycemia [109,110,111,112]. In studies examining the effect of carbohydrates on glucagon secretion, glucagon secretion decreases in response to these stimuli, thereby deeming glucagon redundant for glucose homeostasis during these conditions [113,114]. (Orally administered glucose has, however, been reported to increase plasma concentrations of glucagon in healthy subjects and in patients with type 2 diabetes [115]). This is, however, not the case when high-fat or protein-rich meals are orally administered, as these stimuli increase glucagon secretion [113,116]. In a recent study of the importance of glucagon as a postprandial hormone in healthy individuals, oral administration of whey protein increased both endogenous glucose production and glucose disposal to the same degree [116]. Despite protein ingestion resulting in a concomitant increase in insulin and glucagon concentrations in the blood, euglycemia was preserved throughout the trial [116]. These data suggest that glucagon may be important for glucose flux during the postprandial state after protein ingestion.

A known characteristic of patients with type 2 diabetes is dysregulated glucagon secretion and many have consequently focused on glucagon antagonism [117,118]. Genetic and pharmacologic inhibition of glucagon signaling leads to lowering of fasting plasma glucose concentrations by approximately 1–2 mmol/L in mice and in humans [11,119,120,121,122]. These observations are in line with the early suggestion by Unger and colleagues that increased glucagon signaling is diabetogenic [123,124]. However, administration of a glucagon receptor antagonist (GRA) does not seem to affect postprandial glucose excursions in patients with type 2 diabetes [11], indicating that glucagon is important for glucose homeostasis in the fasting state rather than postprandially. GRAs are currently being pursued as a possible therapeutic target for the treatment of diabetes, though clinical data regarding different antagonists also comprise side effects including increases in liver fat, blood pressure, and low-density lipoprotein (LDL) cholesterol [125]. On the other hand, glucagon agonism has also been suggested as a potential therapeutic strategy for diabetes treatment due to its potential effect on appetite and insulin secretion [126]. Furthermore, co-infusion of glucagon and GLP-1 in overweight humans did not induce hyperglycemia or glucose intolerance but led to a greater decrease in food intake compared to that seen with GLP-1 alone [127]. Additionally, acute glucagon agonism in rodents shows improved glucose tolerance and improved glucose-stimulated insulin secretion after initial glucagon-induced hyperglycemia [126]. These data point to a more complicated interplay between peptide hormones in the regulation of glucose homeostasis than hitherto thought, and several therapeutic strategies involving glucagon signaling are currently being investigated for diabetes treatment. A recent study from our group focused on the pharmacological effects of GRA on postprandial blood glucose in combinatory effect with an insulin receptor antagonist (IRA) [100]. During administration of both GRA and IRA, mice displayed blood glucose responses to a glucose challenge comparable to the vehicle group whereas (and as expected) the blood glucose response to the same challenge was higher after IRA only and lower after GRA only. This further supports the notion that for appropriate glucose homeostasis, both hormones are important [82].

There is no doubt that glucagon is important for the maintenance of normal glucose levels in humans [4]. However, studies carried out during the last decades suggest that glucagon is also a key regulator of hepatic amino acid metabolism, which, as discussed in the following Section, may be linked with hepatic steatosis/fibrosis and potentially dispose towards the development of type 2 diabetes [7].

### 5.2. Glucagon and Amino Acid Metabolism

Glucagon is a powerful stimulus for hepatic amino acid turnover [128,129] by induction of increased activity of enzymes in the urea cycle [130]. Through cAMP-PKA-CREB protein-mediated effects, glucagon regulates several enzymes in the urea cycle at a transcriptional level [129,131]. This is important considering that the capacity of ureagenesis is determined by enzyme activity [130], and long-term regulation of ureagenesis depends on the synthesis rate of the five necessary enzymes (shown in Figure 4). Glucagon also activates the transcription of system A amino acid transporters present in the hepatocyte membrane, thus allowing increased amino acid uptake and substrate availability for ureagenesis [132]. Thus, because ureagenesis is a substrate-regulated pathway, glucagon further induces ureagenesis by providing substrates via increased amino acid uptake. For short-term regulation of ureagenesis, acute activation of at least one of the five enzymes is necessary. *N*-acetyl glutamate (NAG) is the obligatory allosteric activator of carbamoyl phosphate synthetase-1 (CPS-1), and NAG formation, by *N*-acetyl glutamate synthetase (NAGS), is therefore an important factor for short-term regulation of the urea cycle. Steady-state levels of NAG are determined by glutamate and acetyl-CoA concentrations (the substrates for NAG), and activators of NAGS [133]. Glucagon regulates the transcription of NAGS [134], but the rapid increase in NAG concentration induced by glucagon signaling allows rapid activation of CPS-1 and may thus regulate the urea cycle within minutes. Thus, glucagon may acutely regulate hepatic amino acid metabolism via increased ureagenesis [5,128,135,136,137,138,139], and upon pharmacological blockade of glucagon receptor signaling, using a GRA, plasma concentrations of amino acids increase and ureagenesis decrease [135,137,140]. In accordance with this, inhibition of glucagon signaling also reduces the expression of genes involved in hepatic amino acid uptake and metabolism [121,141,142], resulting in hyperaminoacidemia [100,135,140,143]. The rapid changes in urea production upon glucagon signaling cannot be explained by transcriptional changes, and suggest that glucagon also allosterically activates the urea cycle; however, further studies are needed to clarify such a mechanism that may include acetylation [144], phosphorylation [145], sirtuins [146], and/or increased activity of amino acid transporters [121,147]. A detailed map covering the molecular mechanisms of how glucagon receptor signaling may lead to increased hepatic amino acid metabolism remains to be generated.

Secretion of glucagon is powerfully and rapidly stimulated by protein-containing meals [94,148]. Thus, glucagon may serve to regulate postprandial amino acid metabolism in accordance with its acute effect on amino acid metabolism. Postprandial insulin secretion during intake of protein-rich meals also needs to be taken into consideration. Conventionally, the secretion of insulin and glucagon is viewed as being inversely regulated, which is consistent with the opposite effects of the hormones on glucose metabolism. This is, however, not the case upon protein ingestion, as the secretion of both insulin and glucagon increase in response to amino acids [114,116]. To exemplify this further, consider two situations: If only carbohydrates are ingested, plasma concentrations of glucagon will drop in healthy individuals, reaching values close to zero (or to the detection limits of the glucagon assay applied), whereas protein-rich meals are associated with marked increases in glucagon secretion. In the latter situation, the glucagon response is thought to compensate for the potential hypoglycemia induced by an amino acid-driven insulin secretion. However, both mechanisms may also serve to limit amino acid excursions; insulin, by increasing tissue uptake and deposition of amino acids in newly synthesized proteins, and glucagon, by enhancing amino acid metabolism [116,149] (whereas the glucagon receptor is not expressed on human skeletal muscle cells, precluding direct actions of glucagon). In the aforementioned study using both GRA and IRA administration in mice during intravenous administration of amino acids, only glucagon antagonism increased plasma amino acid concentrations compared to vehicle-treated controls, whereas insulin antagonism did not [100]. These findings indicate that clearance of amino acids during the postprandial state depends relatively more on glucagon receptor signaling than on insulin. Whether this also holds true in humans, we do not yet know.

Studies involving genetic or pharmacological ablation of glucagon receptor signaling in rodents are consistently associated with hyperglucagonemia, hyperaminoacidemia, and alpha cell hyperplasia [121,135,150]. In fact, both global and liver-specific elimination of glucagon receptor signaling in mice induces alpha cell hyperplasia. It is especially interesting that hepatic elimination of the glucagon receptor is sufficient to induce an increased alpha cell mass [150], pointing to a direct or indirect signaling loop between the liver and the pancreas. As previously stated, glucagon regulates amino acid metabolism via increased ureagenesis, and when glucagon signaling is inhibited, hyperaminoacidemia occurs. Hyperaminoacidemia is therefore suggested as the factor leading to alpha cell hyperplasia [121,135,141]. This endocrine feedback loop, in which glucagon induces hepatic amino acid metabolism and amino acids in turn stimulate glucagon secretion, has been named the liver–alpha cell axis [7,18]. The glucagonotropic amino acids, alanine and glutamine, may be of particular importance in this feedback loop in mediating alpha cell secretion and growth, respectively [142]. Glucagon receptor blockade also increases the expression of several amino acid transporters in the plasma membrane of alpha cells [121]. Upon pharmacological inhibition of glucagon receptor signaling the pancreatic expression of the amino acid transporter, Slc38a5, increased, and in mice deficient of this amino acid transporter, the development of alpha cell hyperplasia was prevented [121]. These findings did, however, not translate into human islets implanted into mice despite an observed alpha cell proliferation following pharmacological glucagon receptor inhibition [121]. Rather, glucagon receptor inhibition in mice with implanted human islets elicited an increased pancreatic expression of the amino acid transporter Slc38a4 [151]. When investigating molecular mechanisms responsible for increasing alpha cell mass in both rodents and implanted human islets after glucagon receptor blockade, the mechanistic target of the rapamycin (mTOR) pathway has been suggested to be involved [121,151]. However, this is a relatively broad signaling cascade involved in multiple cellular functions including mitosis, and therefore it is of interest to search for potential, more specific downstream targets of amino acid induced alpha cell proliferation.

The aforementioned observations of hyperaminoacidemia, alpha cell hyperplasia, and hyperglucagonemia mimic the manifestations of some metabolic diseases. Both alpha and beta cell proliferation may be increased in islets from patients with recent-onset type 1 diabetes [152]. Furthermore, some, but not all, subjects with type 2 diabetes show elevated glucagon concentrations in the fasting state and sometimes fail to suppress postprandial glucagon secretion [124,153,154]. The idea that not all type 2 diabetes patients display hyperglucagonemia was further nourished when type 2 diabetic patients were compared to patients with non-alcoholic fatty liver disease (NAFLD) [155], showing that NAFLD rather than diabetes was associated with hyperglucagonemia and hyperaminoacidemia [156].

That glucagon is a powerful regulator of amino acid metabolism further becomes evident in cases of extreme glucagon excess and deficiency [41]. Extreme hyperglucagonemia, as observed in patients with glucagon-producing tumors (glucagonomas), is associated with accelerated hepatic amino acid turnover and ureagenesis [90,157,158], resulting in severe hypoaminoacidemia [41,159,160,161] and increased plasma concentrations of urea [143,162]. In studies of the plasma metabolome [163] and proteome [164], the major changes associated with glucagonomas are low levels of plasma amino acids. Glucagonomas are not, as a rule, linked to disturbed glucose metabolism; instead, patients often develop necrolytic migratory erythema, a skin disease associated with severe hypoaminoacidemia [165], which resolves upon intravenous infusions of amino acid [160,166]. Patients with inactivating glucagon receptor mutations (pseudohypoglucagonemia) show hyperaminoacidemia and hyperglucagonemia [167,168] and do not develop severe hypoglycemia. Likewise, pancreatectomized humans show hyperaminoacidemia, which can be normalized by glucagon administration [169]. Collectively, this suggests that one dominant effect of glucagon signaling is both acute (normally postprandial) and chronic regulation of amino acid metabolism; the two may be differentially regulated, and the involved mechanisms, as discussed above, need further exploration.

### 5.3. Glucagon and Lipid Metabolism

The glucagon receptor has, as discussed above, been considered a target for glucose-lowering therapy in type 2 diabetic patients. However, in a phase 2 clinical trial of GRAs (e.g., LY2409021), clinical concerns were raised. Despite the ability of a GRA to effectively lower blood glucose, blocking the glucagon receptor resulted in negative effects on lipid-related processes—the GRA-treated patients had higher total cholesterol, increased hepatic fat fraction (measured by magnetic resonance imaging), and weight gain as compared to controls [16]. Therefore, questions have been raised regarding the relationship between glucagon signaling and lipid metabolism (Figure 5).

Glucagon is known to act mainly on the hepatocytes, which is in line with the fact that the expression levels of glucagon receptors are highest in the liver. Adipocyte expression of the glucagon receptor is possible, but the expression seems to be restricted to rats rather than humans. Therefore, glucagon may primarily regulate lipid metabolism via hepatic signaling. This may involve the following mechanisms. Firstly, when glucagon binds to its receptor on the hepatocyte, it activates cAMP, which in turn accumulates and activates CREB protein. As a consequence, transcription of carnitine acyl transferase (CPT-1) is increased, which enables conversion of fatty acids to acylcarnitines whereby beta-oxidation is activated [93], resulting in the formation of acetate [170,171]. Acetate and CoA react to yield acetyl-coA, which in turn interacts with oxaloacetate to form citrate (thereby inhibiting glycolysis), which enters the citric acid cycle [172]. As a result, hepatic glucagon signaling increases fatty acid catabolism, inhibits glycolysis, and stimulates the citric acid cycle. Secondly, when glucagon binds to its receptor on the hepatocyte, it induces PKA-dependent phosphorylation, which leads to inactivation of acetyl-CoA carboxylase, which functions to catalyze the formation of malonyl-CoA. Because malonyl-CoA is responsible for inhibiting CPT-1, and thus beta-oxidation, lowering malonyl-CoA levels results in a diversion of FFAs to beta-oxidation rather than re-esterification to triglycerols (TGs). Typically, when FFAs are re-esterified, they are stored as TGs and released from hepatocytes into circulation as very-low-density lipoprotein (VLDL). Taken together, glucagon decreases de novo fatty acid synthesis and, consequentially, VLDL release. Lastly, it is possible that glucagon signaling increases the AMP/ATP ratio necessary to activate AMP-activated kinase [173]. This leads to transcriptional activation of peroxisome proliferator-activated receptor-α [2], which induces transcription of beta-oxidation related genes such as CPT-1 and acetyl-CoA oxidase [174,175].

There is additionally in vivo support of the role of glucagon in hepatic lipid metabolism [176]. When mice were administered 30 μg/kg of glucagon acutely, this resulted in decreased FFA and TG plasma concentrations as well as reduced hepatic TG content and secretion. Furthermore, an injection of 10 μg of glucagon every eighth hour for 21 days resulted in a 70% and 38% decrease in plasma concentrations of TG and phospholipids, respectively [177]. In further support of this, synthesis and release of TG were inhibited by glucagon in cultured hepatocytes and in isolated hepatocytes [178,179]. Glucagon also decreased hepatic VLDL synthesis in rats [180]. In humans, hyperglucagonemia (during a pancreatic clamp) resulted in reduced hepatic lipoprotein particle turnover [181], and there has also been indications of a glucagon-mediated increase in hepatic beta-oxidation [182].

In addition to the clinical trials, which demonstrated negative effects on lipid metabolism by GRA administration in humans, similar results have been observed in rodent in vivo studies. In both rats [183] and diabetic (db/db) mice [184], treatment with a glucagon antisense oligonucleotide (impairing glucagon signaling) resulted in increased amounts of hepatic fat. In the livers of global glucagon receptor knockout (Gcgr*^−/−^*) mice, decreased gluconeogenesis and citric acid cycle activity, as well as increased glycolysis, was observed, potentially leading to increased hepatic lipogenesis due to decreased acetyl-CoA oxidation and accumulation. This is supported by reports of upregulation of genes involved in lipogenesis [93,185] and downregulation of enzymes involved in beta-oxidation (e.g., CPT-1) [185]. With regards to the development of steatosis in Gcgr*^−/−^* mice, there are conflicting results [93,186]. When administered a glucagon/GLP-1 receptor co-agonist, type 2 diabetic and obese rodents displayed reduced hepatic steatosis, increased hormone-sensitive lipase (HSL) activity in adipocytes, and improved dyslipidemia [187,188,189,190,191,192,193]. The inhibitory effect on hepatic lipogenesis and the stimulatory effect on beta-oxidation would seem to be mediated particularly by glucagon receptor signaling rather than GLP-1-mediated effects [187,188]. As a result, a number of clinical studies are evaluating the potential for glucagon/GLP-1 receptor co-agonists for obesity and type 2 diabetes treatment [9].

One final aspect regarding the relationship between glucagon signaling and lipid metabolism is the effect in adipocytes. In adipocytes, lipolysis depends on PKA-dependent phosphorylation of HSL [194,195,196] and perilipins, which are present on the surface of lipid droplets [197]. Upon phosphorylation of perilipins, the protein CGI-58 dissociates, which in turn activates adipose triglycerol lipase, which is responsible for converting TGs to diaglycerols. Additionally, phosphorylated perilipins bind HSL and allow it to access the area of the lipid droplet where diacylglycerols are converted to monoglycerols. Monoglycerol lipase hydrolyzes the monoglycerols, resulting in FFAs [198,199,200,201,202] and glycerol, which are released into the circulation. Circulating FFAs and glycerol levels are therefore sensitive indicators of the rate of lipolysis [203]. In rat adipocytes, glucagon receptor mRNA has been detected [204,205], supporting an effect of glucagon on HSL [206,207] and subsequently lipolysis in rat adipocytes [208,209,210,211,212,213,214]. Despite this evidence in rats, there is no clear evidence of expression of a glucagon receptor on human adipocytes, and a lipolytic effect of glucagon at physiological plasma concentrations has been difficult to observe in humans [181,215,216,217,218,219,220,221]. Lipolytic responses to supra-physiological glucagon concentrations have been reported in humans (some resulting in concentrations upwards of >1000 pM) [222,223]. However, this is possibly a result of glucagon-stimulated secretion of catecholamines [224]. The lipolytic effect of supra-physiological glucagon concentration was abolished by insulin in some studies [222,223,225,226,227], and this is consistent with the powerful anti-lipolytic effects of insulin, also found in humans [208,209,225,228,229]. Therefore, if any such lipolytic effect of glucagon on human adipocytes exists, it is only of physiological interest during low insulin secretion [220,228,230].

## 6. Glucagon Resistance and Potential Biomarkers

Does glucagon resistance exist and is it measurable? We hypothesize that glucagon resistance is a molecular phenomenon characterizing impaired physiological effects of glucagon on glucose, amino acid and/or lipid metabolism, which for each of the three listed systems may occur independently. It is important to stress that the hypothesis is new and rests on relatively few observations with limited number of participants. That said, we believe it is important to discuss glucagon resistance as it may aid our understanding of glucagon biology and its contribution to metabolic diseases and potentially allow us to understand the current challenges with glucagon agonism/antagonism by designing targeted therapies like, for example., glucagon-induced hepatic glucose production without tampering with its effect on amino acid and lipid metabolism.

Obesity (and malnutrition) has been suggested as a potential inducer of glucagon resistance in animal models [106,107,231,232], but to what extent this is recapitulated in humans is not yet clear. One study in humans with biopsy-verified hepatic steatosis was presented in 2018 at the American Diabetes Association Scientific Session, reporting that fat infiltration may cause glucagon resistance regarding amino acid metabolism whereas the acute effect of glucagon on hepatic glucose production was preserved [108]. This observation would be in agreement with findings of an attenuated capacity for ureagenesis in individuals with hepatic steatosis [233,234]. One explanation may be that impaired liver function/structure directly affects the transcription of enzymes related to ureagenesis such as CPS-1 or that the effect of glucagon receptor signaling on the same components is impaired. In support of the latter, glucagon receptor knockout (be it liver specific or not) alters the hepatic transcriptome including lowered levels of ureagenesis-related transcripts [7,142,235]. A key regulator of gluconeogenesis is substrate availability, as discussed earlier, and not primarily hormonal systems (insulin and glucagon), whereas ureagenesis may be more dependent on hormonal regulation (but obviously still dependent on a certain flux of substrates) [5,128,138,140]. It will therefore be important to dissect the molecular signals that separate gluconeogenesis from ureagenesis and how these are affected by glucagon receptor signaling. Protein kinase c has been suggested as a potential mediator of glucagon receptor desensitization (measured as impaired production of cAMP, the second-messenger of glucagon receptor activation) [236]. Glucagon receptor internalization [237,238,239] may be a potential mechanism driving glucagon resistance, although others have found glucagon receptor internalization to be minimal compared to, e.g., insulin receptor internalization. It has also been suggested that lipids may interfere with glucagon receptor signaling [81,240] (a case for glucagon resistance was made using adipocytes, relevant for rats [212]). We believe that an important step is to differentiate between chronic (changing protein levels of, e.g., CSP-1) and acute effects of glucagon (e.g., phosphorylation of NAG)—both requires studies examining unbiased changes in the proteome and post-translational modifications, respectively. Further evidence is clearly required to pinpoint impaired glucagon receptor signaling.

How can glucagon resistance be measured and thereby further investigated? In analogy with hepatic insulin resistance, it would be logical to look at effector (insulin/C-peptide) and outcome (glucose) concentrations in plasma. Here, the corresponding parameters would be plasma concentrations of glucagon and amino acids. As discussed above, insulin concentrations might also be considered.

Both measures are sensitive to the accuracy of the measurement methodologies applied (e.g., cross-reactivity to other proglucagon-derived peptides may create false-positive hyperglucagonemia). In addition glucagon levels may be dependent on intact renal function (hyperglucagonemia in patients with renal disease may be composed of not only 1–29 glucagon but also considerable amount of 1–61 [79]). The glucagon/insulin ratio may, on the other hand, be affected by insulin resistance (be it hepatic or peripheral), resulting in increased beta cell secretion (that eventually will fall in line with beta cell failure) and may be invalid in the cases exogenous insulin (late stage type 2 diabetes or type 1 diabetes). That said, to evaluate glucagon’s impact on, for example, glucose production in individuals with and without hepatic steatosis, the manipulation of the glucagon/insulin ratio by somatostatin-induced pancreas clamps and continuous infusion of insulin and glucagon may be a tool to investigate glucagon resistance [241]. Therefore, we speculate that plasma concentrations of glucagon, preferably measured by accurate sandwich ELISA techniques [77], are the golden standard to assess existence of hyperglucagonemia [81] and hence potentially also glucagon resistance. Two recent studies from our group have found support of a glucagon-alanine index that captures the lack of glucagon-induced ureagenesis [155,156], which seems to be specifically impaired in patients with hepatic steatosis and fibrosis [233,234]. Of note, alanine is also involved in the alanine-glucose shuttle between the skeletal muscle and the liver, enabling substrate flow to hepatic gluconeogenesis, and given that glucagon receptor signaling is lacking in skeletal muscle cells (as they do not express the glucagon receptor), the proposed index may therefore reflect the liver–alpha cell axis rather than a muscle–liver–alpha cell axis. However, this will need independent validation by other research groups and larger cohorts (although the index was formed on the basis of results from >1400 individuals).

## 7. Outlook and Conclusions

In this review, we have discussed the essential aspects of glucagon biology with an emphasis on glucagon resistance as a new, potential physiological concept. We provide an overview of the three major biological areas of glucagon receptor signaling: glucose homeostasis, amino acid metabolism, and lipid metabolism. The two latter areas are not as well characterized as the former, and future mechanistic studies involving glucagon agonism/antagonism may be helpful to delineate the physiological importance of glucagon in these areas. Key physiological questions remain unanswered: Are all amino acids affected by glucagon receptor signaling or is it only those with glucagonotropic effects? Does glucagon have a physiological effect on lipid metabolism in humans? The concept of glucagon resistance must be further investigated before conclusions about its relevance in metabolic diseases can be reached. Nevertheless, further studies of the concept will be useful and important for our understanding of glucagon pathobiology.

Research during the coming years may provide answers to whether or not glucagon agonism/antagonism will reach clinical application for treatment of metabolic diseases such as obesity and type 2 diabetes [9]. Interestingly, glucagon agonism (low-dose glucagon) is likely to be applied together with insulin in the dual pump systems development for type 1 diabetes therapy [242]. Furthermore, as glucagon receptor antagonism may result in dyslipidemia, hyperaminoacidemia and hepatic lipid deposition, it has been suggested that glucagon agonism may be of therapeutic relevance for treatment of type 2 diabetes [243]. Future studies will tell whether this holds true, but it seems evident that targeting glucagon receptor signaling alone may be inappropriate because of possible worsening of glycemic control), but perhaps either biased glucagon receptor signaling or glucagon agonism in combination with another therapeutic agent (e.g., GLP-1) may solve this problem.

Thus, glucagon biology remains a challenging but an exciting research object [6].

## Figures and Tables

**Figure 1 ijms-20-03314-f001:**
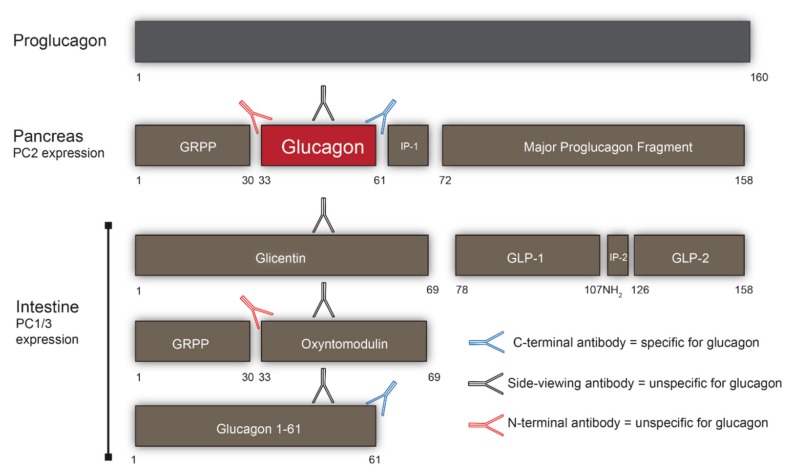
Processing and measurement glucagon. Glucagon (proglucagon 33–61) results from prohormone convertase 2 (PC2)-dependent processing of proglucagon (PG 1–160). In the intestine, PG is processed by prohormone convertase 1/3 (PC1/3) activity to form glicentin (1–69), which may be further cleaved into glicentin-related pancreatic polypeptide (GRPP) and oxyntomodulin (33–69). N-terminal directed antibodies will therefore also cross-react with oxyntomodulin whereas C-terminal antibodies react with proglucagon 1–61, and finally antibodies raised against the mid-region of glucagon will potentially bind to all of the aforementioned peptides. Measurement of glucagon may therefore require a sandwich ELISA targeting both termini.

**Figure 2 ijms-20-03314-f002:**
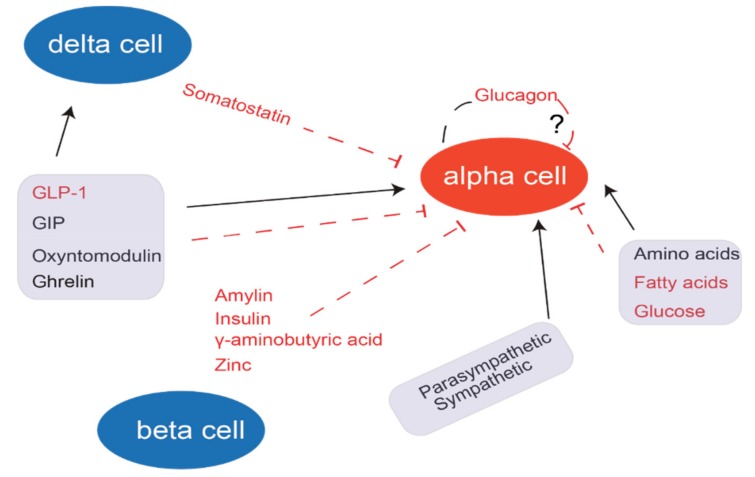
Regulation of glucagon secretion. Several factors regulate the secretion of glucagon; most importantly glucose, amino acids, gastrointestinally derived peptides, the autonomic nervous system (extra-islet regulation), and possibly peptides secreted from the alpha, beta, and delta cells (intra-islet regulation), among which at least the inhibitory action of delta cell-derived somatostatin is well established. Black arrows refer to a stimulatory effect on glucagon secretion and red T bars refer to an inhibitory effect.

**Figure 3 ijms-20-03314-f003:**
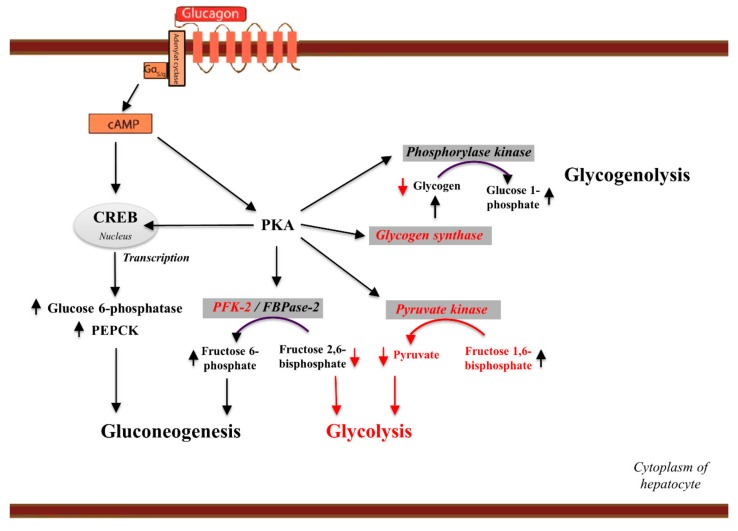
Glucagon effects on hepatic glucose production. Activation of the glucagon receptor results in adenylate cyclase activation and cAMP formation. The increase in intracellular cAMP levels activates protein kinase A (PKA), which phosphorylates the transcription factor cAMP-response-element-binding (CREB) protein. CREB induces the transcription of glucose 6-phosphatase and phosphoenolpyruvate carboxykinase (PEPCK), two enzymes that contribute to increased gluconeogenesis. PKA phosphorylates (hence activates) the phospho-fructokinase 2 (PFK-2)/fructose 2,6-bisphosphatase (FBPase2) protein. Upon phosphorylation, PFK-2 activity is inhibited while FBPase2 activity is activated. Glucagon thus lowers the level of fructose 2,6-bisphosphate and increases fructose 6-phosphate levels, which suppresses glycolysis and increases gluconeogenesis. Secondly, PKA phosphorylates pyruvate kinase, resulting in increased fructose 1,6 bisphosphate levels and decreased pyruvate levels, which leads to reduced glycolysis. Most importantly, PKA phosphorylates phosphorylase kinase, initiating the glycogenolysis cascade increasing the conversion of glycogen to glucose 1-phosphate. Finally, PKA phosphorylates and inhibits glycogen synthase (glucose-6-phosphatase). The red arrows indicate inhibitory actions of glucagon receptor signaling while black arrows indicate stimulatory actions of glucagon receptor signaling.

**Figure 4 ijms-20-03314-f004:**
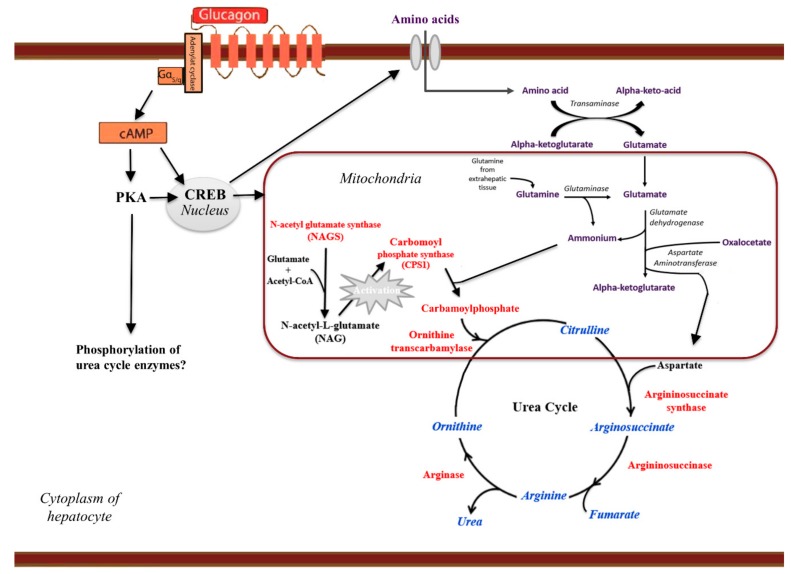
Glucagon’s effect on hepatic amino acid metabolism and ureagenesis. In the hepatocytes, transaminases catalyze the cleavage of the amino group in the form of ammonia, leaving behind the alpha-keto-acid of the respective amino acid. The amino group is transferred to α-ketoglutarate, yielding glutamate. In addition, glutamine is transported via the circulation to the hepatocytes where glutaminase converts glutamine to glutamate and ammonium, which is deposited in the hepatocyte mitochondria. Alanine aminotransferase transfers the amino group from alanine to α-ketoglutarate forming glutamate and leaving pyruvate behind. Glutamate enters the mitochondria of the hepatocyte where glutamate dehydrogenase catalyzes the cleavage of the amino group yielding ammonia and α-ketoglutarate. The ammonium enters the urea cycle by conversion to carbamoyl phosphate, catalyzed by carbamoyl phosphate synthase-1 (CPS-1). Aspartate aminotransferase catalyzes the formation of aspartate via transamination of oxaloacetate and glutamate. Aspartate thus functions as the second nitrogen donor in the urea cycle. The urea cycle starts with N-acetyl-L-glutamate (NAG) formation, which is catalyzed by *N*-acetyl glutamate synthase (NAGS). NAG activates CPS-1, which catalyzes the formation of ammonium to carbamoylphosphate. To form citrulline, the carbamoyl group is transferred from carbamoyl phosphate to ornithine, which is catalyzed by ornithine transcarbamoylase. The next step requires argininosuccinate synthase and argininosuccinase, which convert citrulline and aspartate to arginosuccinate and subsequently to arginine and fumarate. Urea is then produced when arginase cleaves arginine. Glucagon receptor signaling increases urea cycle activity by activating the cAMP-protein kinase A (PKA)-cAMP response element-binding (CREB) protein-pathway, resulting in transcription of urea cycle enzymes and amino acid transporters, the latter serving to increase substrate availability. Glucagon may also allosterically activate the urea cycle by additional PKA mediated phosphorylations (see text).

**Figure 5 ijms-20-03314-f005:**
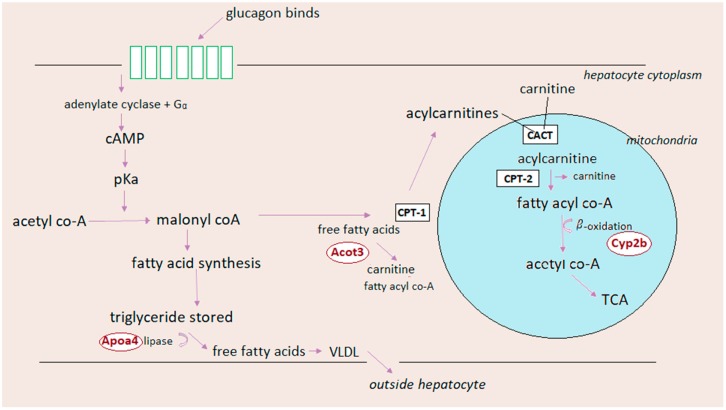
Glucagon effect on lipid metabolism. Activation of the glucagon receptor results in adenylate cyclase-mediated cAMP formation. cAMP accumulation activates cAMP-responsible-binding-protein (CREB), inducing transcription of carnitine acyl transferase-1 (CPT-1) and other genes required for beta-oxidation. CPT-1 promotes conversion of fatty acids to acylcarnitines, which are transported into the mitochondria and broken down to acetate. Acetate and CoA react to form acetyl-CoA, which enters the citric acid cycle. cAMP accumulation activates protein kinase A (PKA), which leads to inactivation of acetyl-CoA carboxylase and thus suppression of malonyl-CoA formation and a disinhibition of beta-oxidation. Thus, glucagon promotes increased beta-oxidation and a decreased fatty acid synthesis and, in turn, very-low-density lipoprotein (VLDL) release.

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
