# Peer review of "Glucagon Receptor Signaling and Glucagon Resistance"

_ijms, 2019, doi:10.3390/ijms20133314_

Round 1
Reviewer 1 Report
The article seems informative and covered many issues relevant to glucagon signaling and resistance. Overall the article is concise and well written. Here are some suggestions to improve further. Although the sections were divided appropriately, I suggest placing section 2-5 in a section and have a focus on signaling related subsection titles as the title of the review is based on signaling. Can also add some information related to delivery systems used in sensitizing or delivering glucagon related compounds in establishing therapeutic effects. for example, ACS Biomater. Sci. Eng.2018, 4, 4225-4235, Nanomaterials 2017, 7, 112 In section 9, only conclusions were given with no clear outlook.
Author Response
We appreciate the time and effort by the reviewer. Thanks for the positive comments.
We have reannotated the sections in the revised manuscript in accordance with your suggestions.
Regarding delivery systems: We do agree this is an interesting topic, although perhaps a bit outside the focus of the current review; so, we expanded the outlook part in section 7 (previously 9).
Reviewer 2 Report
In the review, the authors described literature and their own data focusing on glucagon receptor signaling and glucagon resistance, a critical point in human pathology such as fatty liver disease, obesity and diabetes.
It is important to note that this review is well written, with organized information and clear schemas. Furthermore, this manuscript described precisely the knowledge concerning the biological roles of glucagon in amino acid, lipid and glucose metabolism. In this review, we are able to find critical information concerning roles of glucagon on nutrient metabolism and its integration in different pathologies such as diabetes, obesity and NAFLD.
However, the main and best-described part is mainly the involvement of glucagon in amino-acid metabolism, which is unfortunately not new since authors already nicely described it in the Endocrine review this year (The liver-alpha cell axis; PMID: 30920583).
The most important thing is that there is unfortunately no real and relevant new information concerning the glucagon resistance state, which is actually the main topic of this review except speculations and hypotheses.
Major points:
- The major aim was to describe and bring information concerning a potential glucagon resistance state, as observed for insulin in diabetes. However, this part is clearly shorter and actually need additional information that are more relevant. Authors cannot built review section only based on speculations or hypotheses.
- Looking in the insulin resistance mirror, it would probably be better to assess the glucagon impact directly on hepatocytes (primary culture of hepatocytes from NAFLD or diabetic models) on second messengers (cAMP production as well as PKA and CREB phosphorylation).
Indeed, putative identification of plasma parameters may induce false interpretation or only basic correlation, without any proof of glucagon resistance state.
Minor points:
- I am not sure that utilization of unpublished data bring relevant information especially without peer-review evaluation.
- It appears that authors made error in page 6 (line 37) when they claim that glucose load under OGTT assay led rises of glucagon secretion in both control and T2D subjects, related to Vilsboll group manuscript (reference 116). Indeed, this article has clearly demonstrated that increases of glucagon plasma levels following glucose load were observed only for T2D patients in both oral and IV administration.
Author Response
We appreciate the time and effort by the reviewer and hers/his constructive comments.
First of all, we fully acknowledge that the concept of glucagon resistance is lacking mechanistic studies in humans. The primary main aim of this review is to provide the reader with an overview of the various signaling path of glucagon in regard to glucose, amino acid and lipid metabolism. Regarding the new concept of glucagon resistance, we believe that it is important to discuss this as an inspiration for researchers within the field of glucagon biology and potentially also to think about this when evaluating e.g. glucagon agonism/antagonism in the context of diabetes and fatty liver diseases.
Secondly, we appreciate the point regarding primary human hepatocytes. The entire challenge as we see it is that cAMP response may be sufficient to monitor e.g. glucagon receptor variants but not to reveal whether hepatic actions of glucagon are impaired in hepatocytes from e.g. patients with fatty liver disease. For this, technologies like phosphoproteomics or similar may be necessary to map glucagon receptor signaling apart from the formation of a general second messenger like cAMP. In addition, because of the zonation of the hepatocytes, with e.g. ureagenesis being concentrations in the pericentral zone, it may be difficult to obtain the moist relevant cell types for these experiments. But we agree that this must be done! We have tried to expand this view in the revised manuscript.
The single reference to the unpublished data has been removed and so has the corresponding sentence.
Thanks for noticing the reference to the paper by Vilsboll et al showing hyperglucagonemia during an OGTT. We respectfully disagree with the argument that glucagon did not increase in the healthy subjects. “in patients with type 2 diabetes increasing amounts of oral glucose elicit hypersecretion of glucagon, whereas corresponding IIGIs result in significant glucagon suppression; a phenomenon that is also observed in healthy individuals when larger glucose loads are ingested orally.” – please see the conclusions made in this original study.
Reviewer 3 Report
Congratulations for this excellent piece of work.
I particularly liked the Section on the new concept of hepatic glucagon resistance.
Author Response
Thanks for your positive comments.
Round 2
Reviewer 2 Report
Thanks to the authors for their answers.
Although I do not think that it is crucial to publish several reviews in the same topic during the same year, this review is well written and highlight the glucagon involvement in nutrient metabolism especially amino acid.
Thus I agree that this review now meet the criteria for publication in IJMS journal.